

# Carbon isotope discrimination and the factors affecting it in a summer maize field under different tillage systems

Jichao Cui and Huifang Han

State Key Laboratory of Crop Biology, Key Laboratory of Crop Water Physiology and Drought Tolerance Germplasm Improvement of Ministry of Agriculture, Shandong Agricultural University, Tai'an, China

## ABSTRACT

Based on two years of field experiments, under different soil tillage methods and straw management practices, which included conventional tillage (CT), subsoiling (SS), rotary tillage (RT), and no-tillage (NT), combined with either straw return (S) or straw removal (0), we characterized the dynamic changes in $\Delta^{13}C$ among three height layers [upper (U, 240 cm above the ground), middle (M, 120 cm above the ground), and lower (L, 30 cm above the ground)] of the summer maize canopy. The $\Delta^{13}C$, the factors affecting it, and the relationships between $\Delta^{13}C$ and soil water content (SWC), the leaf area index (LAI), canopy microclimate, and the $CO_2$ concentration were elucidated. The results indicated that the $\Delta^{13}C$ of summer maize at the pre-filling stage was greater than that at the post-filling stage. $\Delta^{13}C$ also varied at different heights, with the order of the $\Delta^{13}C$ values being L > U > M. Among the different tillage methods, the $\Delta^{13}C$ values were ordered $SS_S > CT_S > RT_S > NT_S$. $SS_S$ and $NT_S$ significantly increased the LAI; air temperature and relative humidity tended to gradually decrease with the increase in height of summer maize. Correlation analyses of the various influencing factors and $\Delta^{13}C$ showed that SWC, LAI, air temperature, and $CO_2$ concentration were all positively correlated with $\Delta^{13}C$, in which LAI and air temperature were significantly or extremely significantly positively correlated with $\Delta^{13}C$. In addition, we show that $\Delta^{13}C$ can be used as a prediction index for summer maize yield, providing a theoretical basis for future yield research that may save precious time in summer maize breeding efforts.

Corresponding author
Huifang Han, hhf@sdau.edu.cn

## INTRODUCTION

Grain production in the North China Plain (NCP) accounts for one-third of China's total grain production, which is crucial for ensuring national food security. Maize, which is widely planted in the NCP, is an economically important crop that is an important source of food, forage, and raw material for industrial ethanol production. Long-term use of traditional tillage as the main farming method in this area not only cause soil hardening and shallow plowing layers but also reduces soil water storage and moisture conservation capacity, resulting in adverse consequences for high and stable grain yield (*Bissett et al.,*

2013; *Wang, Sun & Wang, 2018*). The sustainable development of agriculture in the NCP is adversely affected by the destruction of soil structure, which reduces water use efficiency and crop yield (*Balwinder-Singh et al., 2011*; *Latifmanesh et al., 2018*). Therefore, it is vital to find a reasonable land use technique that facilitates the maintenance of soil fertility and the improvement of grain yield. Conservation tillage is one such new land use method that mainly involves less-tillage or no-tillage technology and straw return (*Xue et al., 2019*). A large number of studies have shown that subsoiling, no-tillage, and other conservation tillage measures improve soil structure, enhance resistance to soil erosion and drought, improve soil water storage and moisture conservation capacity, improve water use efficiency (WUE), and significantly increase grain yield (*Jennings et al., 2012*; *Shao et al., 2016*; *Xu et al., 2019*). The quick and accurate evaluation of maize WUE and yield under different tillage methods has important theoretical and practical significance.

Carbon isotopes are natural tracers that indicate changes in carbon processes in agroecosystems and are often used to determine the WUE and yield of ecosystems (*Cui et al., 2009*; *Zhang et al., 2017*). Under different tillage treatments, the photosynthetic carbon sequestration efficiency of summer maize is different due to the differences in plant growth status and the field canopy microenvironment. Stable carbon isotope technology is one of the most effective methods for studying the relationship between plants and the environment. Due to the differences in plant carboxylation efficiency and $^{12}$C and $^{13}$C migration rates in plants and the external environment, the stable carbon isotope ratios differ between plants under different circumstances, which can be used to study ecosystem C cycling and its relationship with the environment as well as to understand changes in ecosystem function. Therefore, it represents an important means of studying ecosystem function and dynamic change. Stable isotope technology can be used to integrate temporal and spatial understanding of ecological processes, indicating the existence of key processes and their long-term development (*Damesin & Lelarge, 2003*). Researchers have applied stable carbon isotope techniques to study agroecosystems, such as to the study of the return of straw as an organic carbon source to cropland, soil effects, and crop photosynthetic carbon interception (*Kristiansen et al., 2005*; *Tharayil et al., 2011*; *Liu et al., 2019a*). *Liu et al. (2019b)* reported that the $\Delta^{13}$C values of wheat organs can be used to evaluate changes and differences in yield and WUE. However, less effort has been directed at understanding the relationship between $\Delta^{13}$C and its influencing factors in summer maize fields with different tillage methods. There are few reports on the effect of canopy microclimate on the carbon isotope changes under different tillage methods.

This study was specifically designed to test the hypothesis that subsoiling and straw return can increase the $\Delta^{13}$C of summer maize, and have a positive correlation between the $\Delta^{13}$C in the middle layer and yield. Moreover, the sensitivity of $\Delta^{13}$C to various influencing factors may be different among the different summer maize layers. The objectives of this study were: (1) to measure the change in $\Delta^{13}$C among the different summer maize layers under different tillage methods; (2) to explore the relationship between $\Delta^{13}$C among the different summer maize layers and its influencing factors. This

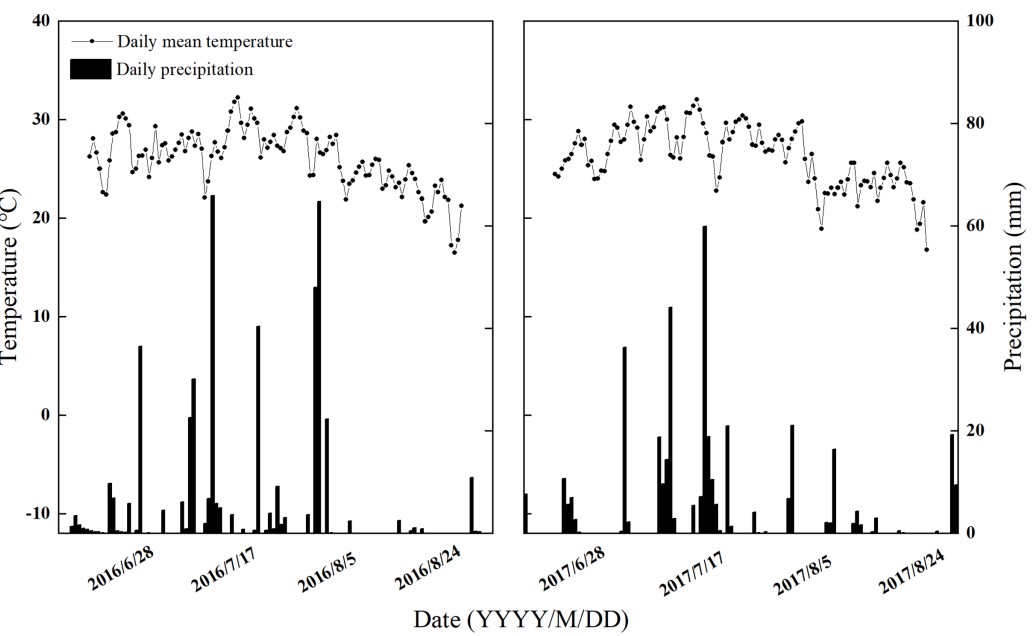

**Figure 1 The daily precipitation and mean temperature at the trial site.** Climate change during the maize growing period from 2016 to 2017 in the experiment.

research can provide a basis for supplementing the carbon sequestration mechanisms of crop plants and for assessing effective soil carbon control and management measures.

# MATERIALS AND METHODS

## Experimental site

The experiment was conducted in a continuous, long-term (14 years) conservation tillage experiment at the Experimental Station of Shandong Agricultural University (117°09′ 13.79″–117°09′12.02″E, 36°09′30.78–36°09′27.59″N); our experiment was performed over 2 years (2016–2017). The experimental area has a typical temperate continental climate with sufficient light and four distinct seasons. The annual average temperature is 13.6 °C. The annual average number of hours of sunlight is 2,462.3 h, and the annual average rainfall is 786.3 mm. This site has a climate that is typical for the NCP. The soil type tested was brown soil with a deep soil layer, and the groundwater level was below 5 m. Climate data for 2016–2017 is shown in Fig. 1. The soil texture is 40% sand, 44% silt, and 16% clay. The major initial properties within the 0–20 cm soil layer were as follows: pH of 7.09, 7.19 g·kg$^{-1}$ SOC, 1.3 g·kg$^{-1}$ total N, 0.79 mg·kg$^{-1}$ available P, and 41.32 mg·kg$^{-1}$ exchangeable K.

## Experimental design

A split-plot design experiment was arranged with three replicates. The plot size was 15 m × 4 m for every replicate. The experiment was divided into the main plot factor: four tillage methods [no-tillage (NT), rotary tillage (RT), subsoiling (SS), and conventional tillage (CT)] and the subplot-factor: straw management [straw return (S) and straw removal (0)].

The winter wheat–summer maize was the typical double cropping rotation system. The winter wheat variety 'Jimai22' was sown in the middle of October each year and harvested in the middle of June the next year. The summer maize variety was 'Zhengdan958', which was sown in the middle and late June of each year and harvested in the first 10 days of October. During the summer maize growth period, basal fertilizer was applied at a rate of 120 kg N ha$^{-1}$, 120 kg P$_2$O$_5$ ha$^{-1}$, and 100 kg K$_2$O ha$^{-1}$ before sowing, and topdressing fertilizer were applied at a rate of 120 kg N ha$^{-1}$ at the large mouth stage.

The four tillage methods in this experiment were carried out only before winter wheat sowing, and summer maize was sowed with no-tillage iron stubble. After harvest, the wheat straw and maize straw from the two seasons were completely crushed (3–5 cm) and returned to the field. For the SS treatment, subsoiler (ZS-180) was used; the soil was plowed for the CT treatment and a rotavator (C250) was used for the RT. Winter wheat was grown using a machine (LXH-150). After the winter wheat was machine-harvested, the straw treatments were applied as described above, and summer maize was sown directly by machine (SHB-2). The field management strategies employed in this experiment were the same as those used in generally high-yielding fields. The specific operating procedures for the tillage treatments could view the supplementary document.

## Test items and methods

In this study, through an ongoing field experiment involving long-term tillage and straw treatments, we measured the $\Delta^{13}C$ of summer maize, soil water content (SWC), leaf area index (LAI), canopy microclimate, and $CO_2$ concentration, and systematically studied the characteristics of $\Delta^{13}C$ among the different summer maize layers and the relationships between $\Delta^{13}C$ and the factors that influence it under different tillage methods.

### Soil water content

Soil water content (SWC), which was measured at depths of 0–10, 10–20, 20–40, and 40–60 cm in the soil profile, was determined by drying method (*Ma et al., 2021*), with three replicates per treatment from the pre-filling and post-filling stages.

### Leaf area index

LAI (m$^2$ m$^{-2}$) was measured at the pre-filling and post-filling stages of summer maize, with three replicates per treatment. Three maize plants were measured for per replicate. The leaf area of a single leaf is equal to the leaf length multiplied by the leaf width multiplied by the leaf coefficient (before flowing stage, unexpanded leaf coefficient is 0.5, expanded leaf coefficient is 0.75; after flowing stage, leaf coefficient is 0.75) (*Zhang et al., 2011*). The LAI was determined by dividing the leaf area in square meters by the ground area in square meters.

### Collection and determination of gas samples

Two maize plants with good growth, basic consistency, and no pest or disease, were covered with a chamber in each plot during the jointing stage. The chamber was designed as a cabinet surrounded by a transparent plastic sheet without covering top to connect to

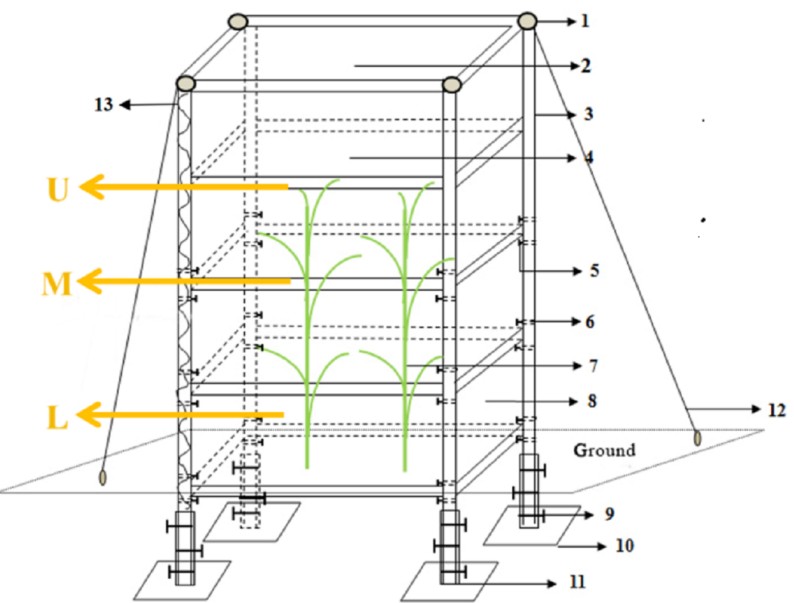

**Figure 2 Simulation test device of microzone.** (1) Iron hoop; (2) top opening indication; (3) 4 cm square tube at side length; (4) PVC film; (5) fixing screw; (6) Screw hole; (7) maize; (8) door; (9) fixed screw on pedestal; (10) iron plate; (11) 5 cm square tube at side length; (12) iron wire; (13) band spring. L (deeper layer, 30 cm above the ground), M (middle layer, 150 cm above the ground), and U (upper layer, 240 cm above the ground).

the atmosphere (Fig. 2), the volume was 2.88 m$^3$ (1.2 m in length, 0.8 m in width, and 3 m in height). The bottom of this chamber was not sealed, embedded into the ground and the joints are sealed with soil. The gas sample was collected by an L-shaped polyvinyl (PVC) pipe from the chamber.

There were divided into three layers [upper layer (U, 240 cm above the ground), middle layer (M, 120 cm above the ground), and lower layer (L, 30 cm above the ground)] to collect the gas samples by a micro-vacuum pump, each sample was packed in 0.5 L gas sampling bag. Then, we took them back to the laboratory, using a Shimadzu gas chromatograph and a stable isotope mass spectrometer to measure the $CO_2$ gas concentrations and $\delta^{13}C$. The measurements were completed within 2 weeks.

### Canopy temperature and relative humidity

The data was collected from 9:00 to 11:00 A.M., at the typical sunny day of the pre-filling and post-filling stages in summer maize. The canopy temperature and relative humidity distribution was measured in the three layers (The height of each layer was the same as the height of collecting the gas sample) by the portable meteorological monitor NK4000. Each layer was randomly repeated 5 times from the central row of the summer maize population.

### Collection and determination of plant samples

Plant samples at L, M, and U were selected at the pre-filling and post-filling stages of summer maize. Plant samples were taken back to the laboratory, dried at 80 °C, and ground.

The $\delta^{13}C$ of the plants was measured using an isotope ratio mass spectrometer with a 0.1–0.2. mg sample.

### Carbon-isotope analysis

Natural $^{13}C$ abundance, expressed in $\delta$ units, can be calculated as follows:

$$\delta^{13}C \ (‰) = (R_{sample}/R_{standard}-1) \times 1000$$

where $R_{sample}$, the isotopic ratio of the study material, and $R_{standard}$, the reference standard.

The source of plant photosynthesis is $CO_2$, and isotopic discrimination values ($\Delta$) can express isotopic effects. Photosynthetic discrimination value can be described as follows:

$$\Delta^{13}C = (\delta^{13}C_a-\delta^{13}C_p)/(1 + \delta^{13}C_p/1000)$$

where $\delta^{13}C_a$ is the $\delta^{13}C$ of atmospheric $CO_2$ and $\delta^{13}C_p$ is the $\delta^{13}C$ of maize plants.

### Grain yield

At harvest, yield was determined in three randomly selected regions of 10 m × 2 rows for each treatment, and each treatment was repeated with three times.

## Statistical analysis

Paired $T$-test and One-way analysis of variance (one-way ANOVA) was performed using SPSS 20.0. Duncan's multiple range test were used to perform multiple comparisons to calculated significant differences at the 5% level among mean value from various groups. Pearson method was used to determine the correlation between difference factors. The test data were processed using Microsoft Excel 2016. Origin 8.0 was used for drawing.

## RESULTS

### Changes in the $\Delta^{13}C$ of summer maize

In both growing seasons, the $\Delta^{13}C$ among the different summer maize layers was significantly influenced by both tillage methods and straw management (Table 1). Different straw management methods resulted in different $\Delta^{13}C$ values, and straw return significantly increased $\Delta^{13}C$. Regardless of layer and growing seasons, the average $\Delta^{13}C$ (5.21‰) under straw return was significantly higher than that under straw removal (5.09‰). The results showed that the fluctuations in $\Delta^{13}C$ in summer maize were consistent across the four tillage methods. $SS_S$ significantly increased the $\Delta^{13}C$. The $\Delta^{13}C$ performance was ranked $SS_S > CT_S > RT_S > NT_S$, and there were significant differences in $\Delta^{13}C$ under different tillage methods, which indicated that the tillage methods significantly affected the $\Delta^{13}C$ of summer maize.

At different growth stages, the $\Delta^{13}C$ among the different summer maize layers showed that the $\Delta^{13}C$ at the pre-filling stage was higher than that at the post-filling stage. The lowest value of $\Delta^{13}C$ was in M, and the order of $\Delta^{13}C$ among the different summer maize layers was L > U > M.

**Table 1 Vertical distribution of the $\Delta^{13}C$ in summer maize canopy at pre-filling and post-filling under different treatments.** $CT_0$ (conventional tillage with straw removal), $SS_0$ (subsoiling with straw removal), $RT_0$ (rotary tillage with straw removal), $NT_0$ (no-tillage with straw removal), $CT_S$ (conventional tillage with straw return), $SS_S$ (subsoiling with straw return), $RT_S$ (rotary tillage with straw return) and $NT_S$ (no-tillage with straw return). L (lower layer, 30 cm above the ground), M (middle layer, 150 cm above the ground), and U (upper layer, 240 cm above the ground). Different letters in each column indicate significant differences between different treatments ($P < 0.05$; Duncan's test).

| Treatments | Pre-filling (‰) | | | Post-filling (‰) | | | Pre-filling (‰) | | | Post-filling(‰) | | |
|---|---|---|---|---|---|---|---|---|---|---|---|---|
| | L | M | U | L | M | U | L | M | U | L | M | U |
| | 2016 | | | | | | 2017 | | | | | |
| Straw | | | | | | | | | | | | |
| 0 | 5.40 ± 0.19a | 4.91 ± 0.16a | 5.31 ± 0.17a | 5.28 ± 0.18a | 4.90 ± 0.14a | 5.24 ± 0.18a | 5.29 ± 0.12b | 4.85 ± 0.12b | 5.16 ± 0.10b | 5.14 ± 0.11a | 4.65 ± 0.15a | 4.96 ± 0.11b |
| S | 5.50 ± 0.20a | 5.05 ± 0.19a | 5.40 ± 0.17a | 5.39 ± 0.18a | 4.97 ± 0.18a | 5.32 ± 0.16a | 5.47 ± 0.10a | 5.06 ± 0.17a | 5.31 ± 0.17a | 5.20 ± 0.10a | 4.74 ± 0.18a | 5.06 ± 0.11a |
| Tillage | | | | | | | | | | | | |
| CT | 5.51 ± 0.08b | 5.05 ± 0.12b | 5.40 ± 0.06b | 5.37 ± 0.10b | 5.00 ± 0.07b | 5.30 ± 0.07b | 5.41 ± 0.11ab | 5.02 ± 0.15a | 5.31 ± 0.12ab | 5.21 ± 0.08ab | 4.75 ± 0.08b | 5.03 ± 0.10ab |
| SS | 5.71 ± 0.08a | 5.19 ± 0.10a | 5.57 ± 0.09a | 5.57 ± 0.09a | 5.11 ± 0.13a | 5.49 ± 0.08a | 5.51 ± 0.10a | 5.09 ± 0.16a | 5.37 ± 0.14a | 5.26 ± 0.07a | 4.88 ± 0.13a | 5.13 ± 0.08a |
| RT | 5.35 ± 0.11c | 4.89 ± 0.12c | 5.28 ± 0.09c | 5.25 ± 0.08c | 4.90 ± 0.04b | 5.26 ± 0.06b | 5.34 ± 0.12b | 4.92 ± 0.15ab | 5.20 ± 0.10bc | 5.15 ± 0.09bc | 4.61 ± 0.11c | 4.96 ± 0.12b |
| NT | 5.23 ± 0.07d | 4.78 ± 0.07c | 5.16 ± 0.07d | 5.14 ± 0.08d | 4.73 ± 0.06c | 5.07 ± 0.11c | 5.27 ± 0.13b | 4.79 ± 0.10b | 5.07 ± 0.07c | 5.06 ± 0.10c | 4.54 ± 0.08c | 4.92 ± 0.10b |
| Coupling | | | | | | | | | | | | |
| $CT_0$ | 5.46 ± 0.04cd | 4.96 ± 0.07c | 5.36 ± 0.05c | 5.32 ± 0.08c | 4.97 ± 0.05bc | 5.28 ± 0.08c | 5.33 ± 0.07cd | 4.89 ± 0.07cd | 5.21 ± 0.04bc | 5.19 ± 0.10ab | 4.73 ± 0.07bc | 4.98 ± 0.11bc |
| $SS_0$ | 5.66 ± 0.07ab | 5.11 ± 0.07b | 5.52 ± 0.07ab | 5.51 ± 0.07b | 5.02 ± 0.14b | 5.44 ± 0.09ab | 5.43 ± 0.05bc | 4.96 ± 0.07cd | 5.25 ± 0.05b | 5.22 ± 0.05ab | 4.81 ± 0.07ab | 5.07 ± 0.05ab |
| $RT_0$ | 5.28 ± 0.08e | 4.80 ± 0.09d | 5.22 ± 0.09de | 5.19 ± 0.05de | 4.88 ± 0.04c | 5.24 ± 0.09cd | 5.24 ± 0.05de | 4.83 ± 0.12de | 5.13 ± 0.07bcd | 5.13 ± 0.07bc | 4.55 ± 0.07cd | 4.91 ± 0.11bc |
| $NT_0$ | 5.20 ± 0.08e | 4.75 ± 0.06d | 5.13 ± 0.07e | 5.08 ± 0.05e | 4.71 ± 0.07d | 5.01 ± 0.08e | 5.16 ± 0.08e | 4.72 ± 0.06e | 5.04 ± 0.05d | 5.01 ± 0.09c | 4.50 ± 0.07d | 4.86 ± 0.09c |
| $CT_S$ | 5.56 ± 0.07bc | 5.13 ± 0.11b | 5.44 ± 0.05bc | 5.41 ± 0.10bc | 5.03 ± 0.09b | 5.32 ± 0.07bc | 5.49 ± 0.06ab | 5.15 ± 0.05ab | 5.41 ± 0.08a | 5.22 ± 0.07ab | 4.77 ± 0.11ab | 5.07 ± 0.07ab |
| $SS_S$ | 5.75 ± 0.06a | 5.27 ± 0.06a | 5.61 ± 0.10a | 5.63 ± 0.05a | 5.19 ± 0.06a | 5.54 ± 0.04a | 5.59 ± 0.06a | 5.22 ± 0.08a | 5.48 ± 0.08a | 5.30 ± 0.06a | 4.94 ± 0.15a | 5.19 ± 0.07a |
| $RT_S$ | 5.42 ± 0.09d | 4.98 ± 0.06c | 5.34 ± 0.05cd | 5.31 ± 0.05cd | 4.92 ± 0.04bc | 5.27 ± 0.04c | 5.43 ± 0.06bc | 5.01 ± 0.12bc | 5.26 ± 0.10b | 5.16 ± 0.12abc | 4.66 ± 0.13bcd | 5.01 ± 0.12bc |
| $NT_S$ | 5.25 ± 0.06e | 4.81 ± 0.07d | 5.19 ± 0.05e | 5.19 ± 0.05de | 4.75 ± 0.03d | 5.13 ± 0.10de | 5.37 ± 0.07bc | 4.85 ± 0.08de | 5.09 ± 0.10cd | 5.11 ± 0.09bc | 4.57 ± 0.09cd | 4.98 ± 0.08bc |
| Interaction | | | | | | | | | | | | |
| Straw 秸秆 | 0.004 | 0.000 | 0.007 | 0.001 | 0.021 | 0.033 | 0.000 | 0.000 | 0.000 | 0.096 | 0.045 | 0.011 |
| Tillage | 0.000 | 0.000 | 0.000 | 0.000 | 0.000 | 0.000 | 0.000 | 0.000 | 0.000 | 0.005 | 0.000 | 0.006 |
| Straw × Tillage | 0.741 | 0.500 | 0.899 | 0.977 | 0.396 | 0.683 | 0.885 | 0.491 | 0.193 | 0.843 | 0.859 | 0.990 |

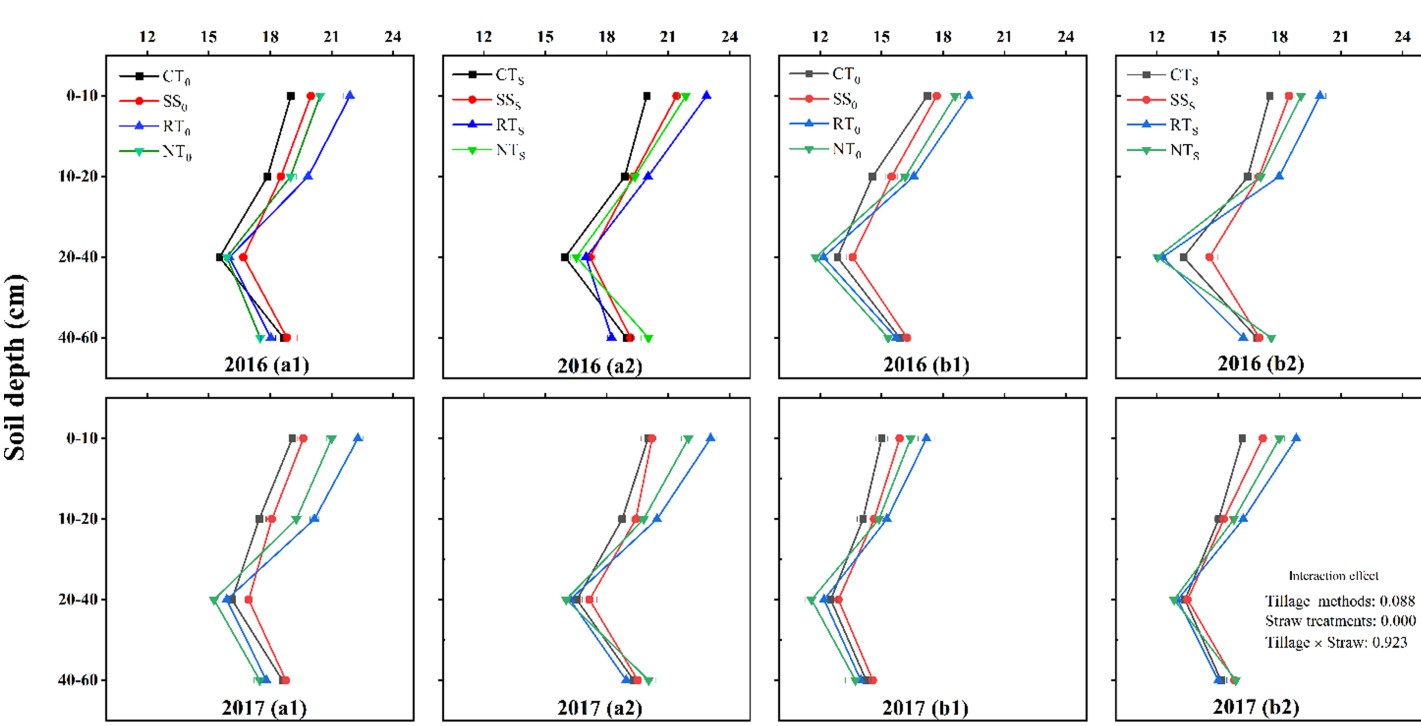

**Figure 3** **Effects of tillage methods on soil water content.** a1 (straw removal at the pre-filling), a2 (straw return at the pre-filling), b1 (straw removal at the post-filling), b2 (straw return at the post-filling). $CT_0$ (conventional tillage with straw removal), $SS_0$ (subsoiling with straw removal), $RT_0$ (rotary tillage with straw removal), $NT_0$ (no-tillage with straw removal), $CT_S$ (conventional tillage with straw return), $SS_S$ (subsoiling with straw return), $RT_S$ (rotary tillage with straw return) and $NT_S$ (no-tillage with straw return). Horizontal bars are standard errors.

## Analysis of the factors affecting the $\Delta^{13}C$ of summer maize

### Soil water content

The SWC in the summer maize field decreased seasonally as the rainfall intensity decreased, and the trends for changes in SWC with time varied slightly between the 2 years of our study due to differences in the rainfall distributions between the 2 years (Figs. 1 and 3). The SWC at the pre-filling stage was higher than that at the post-filling stage. The SWC under the condition of straw return was higher than that under straw removal, which indicated that straw return enhanced soil water holding capacity, especially in the 0–10 cm soil layer. In the 0–10 cm soil layer, the 2 years' average SWC of $CT_S$, $SS_S$, $RT_S$, and $NT_S$ was increased by 4.82%, 5.68%, 5.30%, and 5.91%, respectively, under straw return, compared with the straw removal treatments.

Across the soil depth range of 0–60 cm, SWC decreased initially and then increased with increasing soil depth. In the 0–20 cm soil layer, the SWC was significantly lower under CTs than under the other treatments with straw return, and the SWC order was $RT_S$ > $NT_S$ > $SS_S$ > $CT_S$. In the 20–40 cm soil layer, the SWC of $SS_S$ was significantly higher than that of the other treatments, and the SWC of $NT_S$ was the lowest among the different tillage treatments. In the 40–60 cm soil layer, the SWC performance ranking of the different tillage methods under straw return was $NT_S$ > $SS_S$ > $CT_S$ > $RT_S$.
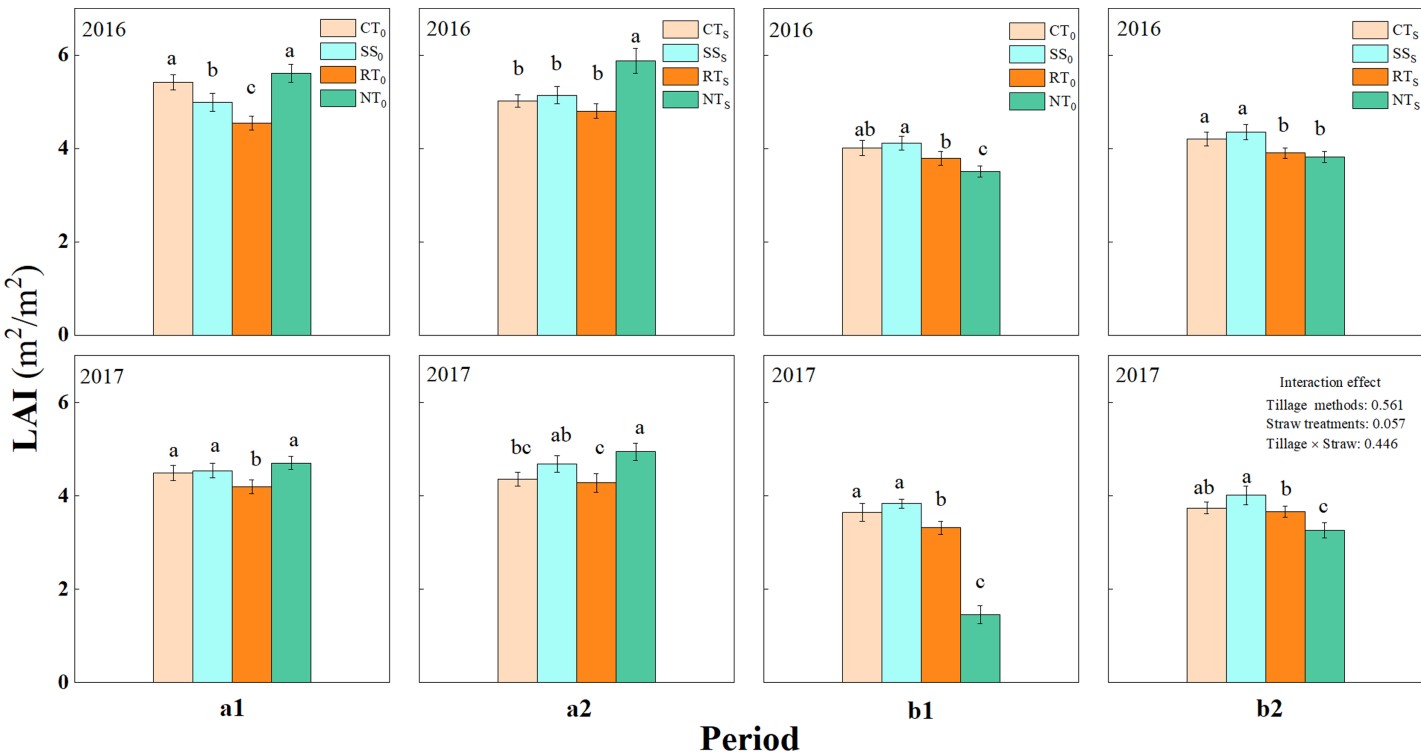

**Figure 4 Leaf area index at the pre-filling and the post-filling.** a1 (straw removal at the pre-filling), a2 (straw return at the pre-filling), b1 (straw removal at the post-filling), b2 (straw return at the post-filling). $CT_0$ (conventional tillage with straw removal), $SS_0$ (subsoiling with straw removal), $RT_0$ (rotary tillage with straw removal), $NT_0$ (no-tillage with straw removal), $CT_S$ (conventional tillage with straw return), $SS_S$ (subsoiling with straw return), $RT_S$ (rotary tillage with straw return) and $NT_S$ (no-tillage with straw return). Different letters in each column indicate significant differences between different treatments ($P < 0.05$; Duncan's test). The graphs in upper and lower panel represent data from 2016 and 2017, respectively.

### Leaf area index

The LAI of summer maize at the pre-filling stage was greater than that at the post-filling stage (Fig. 4). Overall, the LAI of the straw return treatment was greater than that of the straw removal treatment. At the pre-filling stage, LAI for $SS_S$, $RT_S$, and $NT_S$ were higher than those for $SS_0$, $RT_0$, and $NT_0$, except for $CT_S$. Under straw return conditions, the order of the LAI of summer maize was $NT_S > SS_S > CT_S > RT_S$ at the pre-filling stage, and it was $SS_S > CT_S > RT_S > NT_S$ at the post-filling stage.

Under straw return conditions, the LAI of the $NT_S$ treatment was significantly higher than that of other treatments at the pre-filling stage. However, at the post-filling stage, the LAI of $NT_S$ treatment decreased and was the lowest among the tillage treatments. The average difference between the post-filling and pre-filling LAI for $CT_S$, $SS_S$, $RT_S$, and $NT_S$ was 0.72, 0.73, 0.75, and 1.87, respectively. Hence, the LAI of $NT_S$ decreased the most between pre-filling and post-filling, indicating that the leaf senescence rate under the $NT_S$ treatment was higher than those under the other three tillage methods.

### Air temperature in the summer maize canopy

The changes in air temperature among the different summer maize layers at the pre-filling and the post-filling stages for summer maize in 2016 and 2017 are shown in Fig. 5.

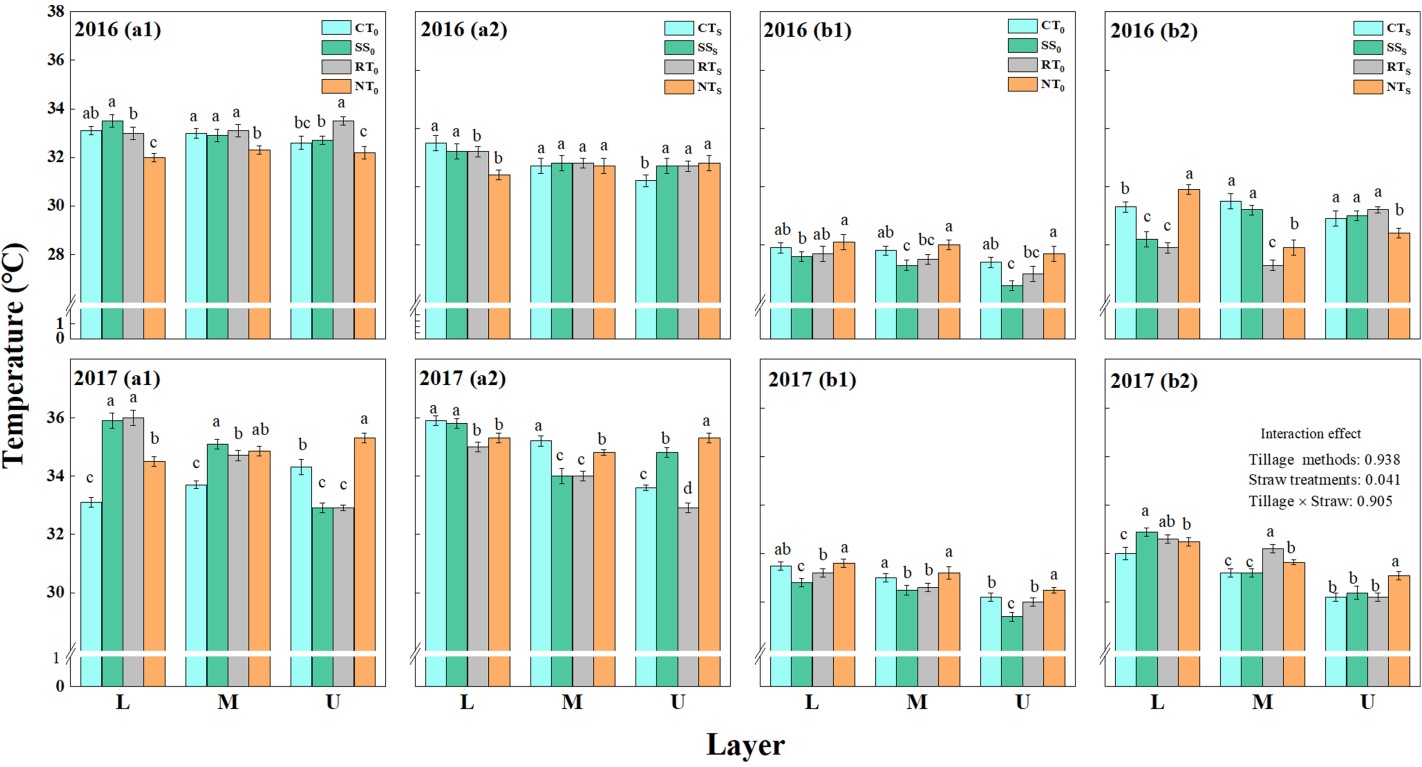

**Figure 5 Vertical distribution of air temperature in summer maize canopy.** a1 (straw removal at the pre-filling), a2 (straw return at the pre-filling), b1 (straw removal at the post-filling), b2 (straw return at the post-filling). $CT_0$ (conventional tillage with straw removal), $SS_0$ (subsoiling with straw removal), $RT_0$ (rotary tillage with straw removal), $NT_0$ (no-tillage with straw removal), $CT_S$ (conventional tillage with straw return), $SS_S$ (subsoiling with straw return), $RT_S$ (rotary tillage with straw return) and $NT_S$ (no-tillage with straw return). EFS (early filling stage) and LFS (last filling stage). L (lower layer, 30 cm above the ground), M (middle layer, 150 cm above the ground), and U (upper layer, 240 cm above the ground). Different letters in each column indicate significant differences between different treatments ($P < 0.05$; Duncan's test).

The air temperature decreased with the increase in height, and the air temperature at the pre-filling stage was significantly higher than that at the post-filling stage (both straw removal and straw return treatments). The air temperature under straw return was significantly higher than that under straw removal, which indicated that straw return significantly affected air temperature.

At the pre-filling stage, the air temperature of $CT_S$ was the highest in the lower layers of the summer maize canopy, whereas the highest temperature under $NT_S$ was observed in the upper layer. At the post-filling stage, the air temperature of $NT_S$ was highest among the three layers under straw removal in both two growing seasons, whereas $SS_S$ was lowest.

### Relative humidity in the summer maize canopy

Straw return decreased relative humidity, and the relative humidity was higher at the pre-filling stage than at the post-filling stage (Fig. 6). Relative humidity was higher in the lower layer of the summer maize canopy than it was in the middle and upper layers. The trends for the effects on relative humidity were similar among the different tillage methods and straw management conditions, with a high level of influence apparent in the lower layer.

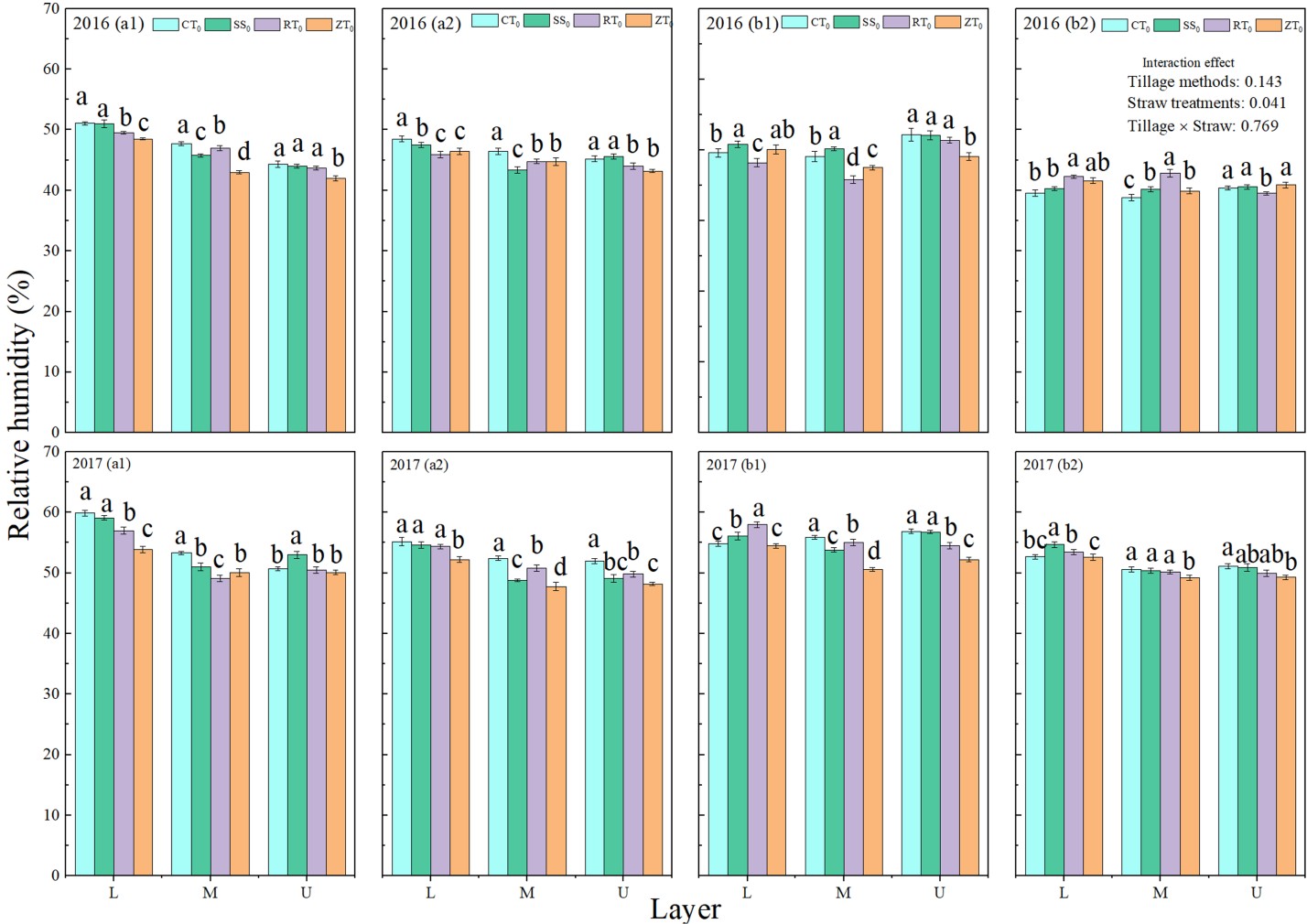

**Figure 6 Vertical distribution of relative humidity in summer maize canopy.** a1 (straw removal at the pre-filling), a2 (straw return at the pre-filling), b1 (straw removal at the post-filling), b2 (straw return at the post-filling). $CT_0$ (conventional tillage with straw removal), $SS_0$ (subsoiling with straw removal), $RT_0$ (rotary tillage with straw removal), $NT_0$ (no-tillage with straw removal), $CT_S$ (conventional tillage with straw return), $SS_S$ (subsoiling with straw return), $RT_S$ (rotary tillage with straw return) and $NT_S$ (no-tillage with straw return). EFS (early filling stage) and LFS (last filling stage). L (lower layer, 30 cm above the ground), M (middle layer, 150 cm above the ground), and U (upper layer, 240 cm above the ground). Different letters in each column indicate significant differences between different treatments ($P < 0.05$; Duncan's test).

At the pre-filling stage, the relative humidity under straw return was lower than that under straw removal. The results showed that the relative humidity of $NT_S$ treatment was significantly lower than those of the $CT_S$, $SS_S$, and $RT_S$ treatments in 2016 growing seasons under straw at the pre-filling stage. The average humidity in the $CT_S$, $SS_S$, $RT_S$, and $ZT_S$ treatments was 49.96%, 48.16%, 48.28%, and 47.13%, respectively, which was 1.06%, 5.36%, 2.41%, and 1.67% lower, respectively, than that of average humidity under straw removal, indicating that straw return reduced soil water evaporative loss. Straw return and no-tillage treatments had high water retention, so the water dispersion under those treatments was lower during water shortages, and the relative humidity was low. In conclusion, the relative humidity of $CT_0$ was the highest, and $NT_S$ was the lowest. Under

**Table 2 Correlation coefficients of $\Delta^{13}C$ and its affecting factors under different treatments.** L (lower layer, 30 cm above the ground), M (middle layer, 150 cm above the ground), and U (upper layer, 240 cm above the ground). SWC60, LAI, T and M represented soil water content at 0–60 cm soil layer, leaf area index for summer maize, air temperature in the summer maize canopy, and relative humidity in the summer maize canopy.

| Layer | Parameter | $\Delta^{13}C$ | $SWC_{60}$ | LAI | T | M | $CO_2$ |
|-------|-----------|------|--------|-----|---|---|-----|
|   | $SWC_{60}$ | 0.547** | 1 | | | | |
|   | LAI | 0.577** | 0.759** | 1 | | | |
| L | T | 0.449* | 0.871** | 0.552** | 1 | | |
|   | M | −0.108 | 0.125 | −0.062 | 0.393* | 1 | |
|   | $CO_2$ | 0.261 | 0.271 | 0.430* | −0.062 | −0.719** | 1 |
|   | $SWC_{60}$ | 0.513** | 1 | | | | |
|   | LAI | 0.554** | 0.759** | 1 | | | |
| M | T | 0.437* | 0.895** | 0.631** | 1 | | |
|   | M | −0.282 | −0.039 | −0.186 | 0.137 | 1 | |
|   | $CO_2$ | 0.247 | 0.127 | 0.327 | −0.143 | −0.829** | 1 |
|   | $SWC_{60}$ | 0.522** | 1 | | | | |
|   | LAI | 0.582** | 0.759** | 1 | | | |
| U | T | 0.423* | 0.907** | 0.670** | 1 | | |
|   | M | −0.440* | −0.193 | −0.308 | −0.140 | 1 | |
|   | $CO_2$ | 0.390* | 0.150 | 0.339 | 0.021 | −0.850** | 1 |

Notes:
* Significance ($P < 0.05$).
** Significance ($P < 0.01$).

the different tillage methods, straw return increased relative humidity in the summer maize canopy.

## Analysis of $\Delta^{13}C$ and its influencing factors

SWC, LAI, air temperature, relative humidity, and the $CO_2$ concentration can directly or indirectly affect the $\Delta^{13}C$ of summer maize (Table 2). The results showed that the correlations of SWC, LAI, air temperature, relative humidity, and the $CO_2$ concentration with $\Delta^{13}C$ were significantly different among the different layers of summer maize. Among them, SWC, LAI, and air temperature were significantly positively correlated with the $\Delta^{13}C$ in the lower layer, and, for SWC and LAI, this correlation was highly significant. By contrast, the relative humidity and the $CO_2$ concentration had no significant correlation with $\Delta^{13}C$. In the middle layer, SWC, LAI, and air temperature significantly affected the $\Delta^{13}C$ of summer maize, among which SWC and LAI with $\Delta^{13}C$ had a very significant positive correlation. In the upper layer, the $CO_2$ concentration and air temperature had significant positive correlations with $\Delta^{13}C$, and SWC and LAI had an extremely significant positive correlation with $\Delta^{13}C$, whereas relative humidity had a very significant negative correlation with $\Delta^{13}C$. In addition, SWC and LAI were extremely significantly positively correlated with air temperature under the different summer maize layers, which indicated that the increases in SWC and LAI had a beneficial effect on the increase in air temperature, and they synergistically promoted or inhibited changes in air

temperature, thereby affecting $\Delta^{13}C$; SWC and LAI had an extremely significant positive correlation, and the larger the SWC was, the larger the LAI was, indicating that water availability significantly affected the LAI of summer maize plants and promoted the growth of summer maize plants under good soil moisture conditions, thereby increasing the LAI. There was a significant positive correlation between the $CO_2$ concentration and LAI in the lower layer, and an extremely significant negative correlation between the $CO_2$ concentration and relative humidity.

According to the correlation coefficient, the rankings for the factors affecting the $\Delta^{13}C$ of summer maize canopy in the lower, middle, and upper layers was LAI > SWC > air temperature > $CO_2$ concentration > relative humidity, LAI > SWC > air temperature > relative humidity > $CO_2$ concentration, and LAI > SWC > relative humidity > air temperature > $CO_2$ concentration, respectively. Under the different summer maize layers, various factors have different contributions to $\Delta^{13}C$. SWC, LAI, and air temperature were significantly or extremely significantly correlated with the $\Delta^{13}C$ in L, M, and U, all of which were the main factors affecting $\Delta^{13}C$, and the $CO_2$ concentration played an important role in $\Delta^{13}C$ in U. Interestingly, we found that there was a negative correlation between relative humidity and $\Delta^{13}C$ at different canopy heights in summer maize, among which, in U, relative humidity and $\Delta^{13}C$ had a significant negative correlation.

### The relationship between the $\Delta^{13}C$ of summer maize and yield

The results of the correlation analysis for summer maize $\Delta^{13}C$ and yield under both straw removal and straw return treatments is shown in Fig. 7. It can be seen that the $\Delta^{13}C$ of summer maize at the pre-filling stage showed a significant positive correlation with the summer maize yield. The $\Delta^{13}C$ of summer maize in the middle of the canopy had a very significant positive correlation with the summer maize yield, with a significance level reaching 0.0001. However, there was no significant correlation between $\Delta^{13}C$ and yield at the post-filling stage in summer maize.

## DISCUSSION

### The $\Delta^{13}C$ at different layers in the summer maize canopy

Temperature, moisture, and the atmospheric $CO_2$ concentration all affect the plant leaf $\Delta^{13}C$ value (*Li et al., 2017*). Under different tillage systems, $\Delta^{13}C$ at the pre-filling stage was greater than that at the post-filling stage, and the plants fractionated heavy isotopes more vigorously at the pre-filling stage, that is, their ability to distinguish $^{13}CO_2$ was stronger. This was due to the senescence of summer maize plants at the post-filling stage and the accompanying waning of the photosynthetic assimilation capacities of the summer maize plants. The $\Delta^{13}C$ values were significantly different among the different summer maize layers. The lowest value of $\Delta^{13}C$ was found in the middle of the summer maize canopy, whereas the highest $\Delta^{13}C$ was in the lower maize canopy. The $\Delta^{13}C$ order among the different summer maize layers was L > U > M. There may be two main reasons: soil respiration emits large quantities of $CO_2$ to the atmosphere, resulting in a lower $\delta^{13}C$ near the ground atmosphere. *Liu et al. (2019c)* suggested that the contribution of summer maize field soil emissions to photosynthesis is 20.37–29.03%. As the canopy

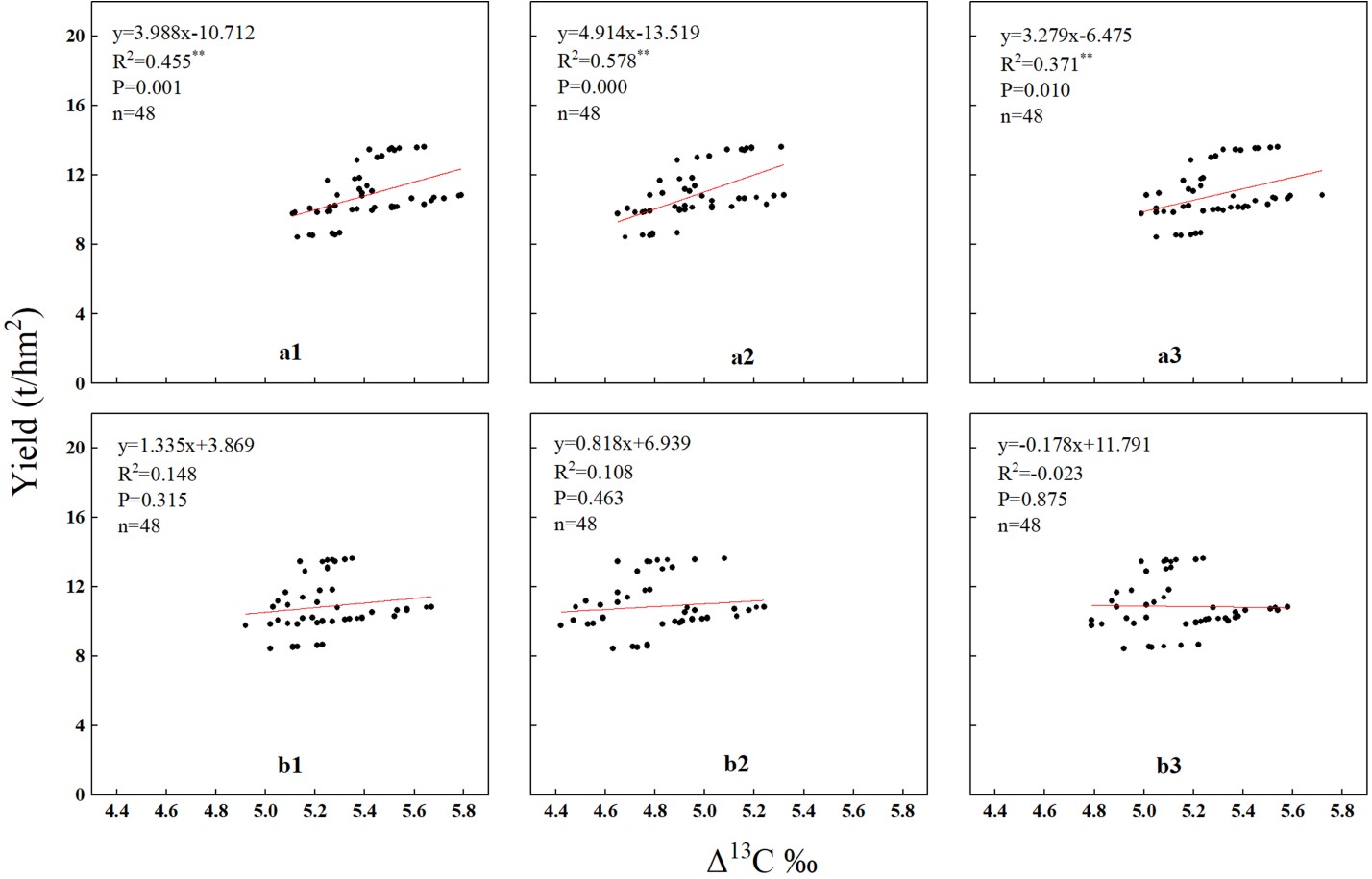

**Figure 7 The relationship between the $\Delta^{13}$C of summer maize and yield.** a1 (L at the pre-filling), a2 (M at the pre-filling), a3 (U at the pre-filling), b1 (L at the post-filling), b2 (M at the post-filling), b3 (U at the post-filling). "t/hm$^2$" means tonnes/hectare.

height increases, the ratio of the $CO_2$ released from the soil to atmospheric $CO_2$ gradually declines. To a certain extent, increasing soil carbon emissions can increase the rate of maize photosynthesis. Plant photosynthetic carbon assimilation is discriminatory, and its reaction matrix is mainly $^{12}CO_2$. Similarly, if respiration releases more $CO_2$, predominantly containing $^{12}$C, the $\delta^{13}$C of soil $CO_2$ will be lower than that of the plant canopy; that is, the $\Delta^{13}$C will be greater in the lower layer than in the middle layer, as it was in our study. Secondly, due to photosynthesis, the canopy took the lead in assimilating $^{12}CO_2$, which gradually increased the $\delta^{13}$C value of the canopy $CO_2$. The upper canopy leaves had a large light-receiving area and were distant from the source of $CO_2$ generated by soil respiration (*i.e.*, the ground). This is consistent with the research results of *Silveira et al. (1989)* that the forest canopy has obvious stratification phenomenon, and the closer to the forest surface, the more obvious the isotope depletion of plant leaves.

In the summer maize growing season, the $\Delta^{13}$C values of $SS_S$, $CT_S$, and $RT_S$ were significantly higher than that of $NT_S$. The main reasons may be as follows: first, the degree of soil disturbance under $SS_S$, $CT_S$, and $RT_S$ was higher than that under $NT_S$. During the summer maize growing season, the temperature was higher, the rainfall was frequent,

so the soil temperature and soil moisture were also high, which provided the theoretical conditions for soil respiration. Soil respiration releases a large amount of $CO_2$. Second, tillage improves soil permeability, enhances the migration and diffusion of gases in the soil, promotes the decomposition of organic matter, and increases the release of $CO_2$ from soil respiration. Compared with the other tillage methods, $NT_S$ effectively reduced soil disturbance and the chance of soil-air contact, reduced the decomposition of soil organic matter, and significantly reduced soil respiration, which was consistent with previous research results (*Six et al., 2004*; *Yonemura et al., 2014*). Different straw management conditions resulted in different $\Delta^{13}C$ values. In the two periods, the $\Delta^{13}C$ of the straw return treatment was significantly higher than that of the straw removal treatment. The main reasons may be as follows: firstly, the plant straw that was returned to the soil is a carbon source, and the straw was decomposed into various organic and inorganic substances, providing many nutrients for plant growth and releasing $CO_2$ (*Negassa et al., 2015*; *Song et al., 2017*); secondly, straw return improved the physical and chemical properties of soil, maintained soil structure and fertility (*Jones et al., 2005*), promoted the growth and extension of plant roots, and promoted the growth of the aboveground plant parts, increased the LAI, and improved photosynthesis. The increase in photosynthesis increased the capture of $CO_2$, and finally affected the $\Delta^{13}C$. The lower $\Delta^{13}C$ for summer maize may reduce lower photosynthesis resulted lower yield. And *Wei et al. (2019)* found that the more $^{13}C$-photosynthate distribution to ears, and the less $^{13}C$-photosynthate distribution to stems could achieve a greater yield consequently. In future studies, we will investigate whether the decrease of $\Delta^{13}C$ reduced photosynthesis for summer maize.

## Influencing factors of $\Delta^{13}C$

Among the five factors (*i.e.*, SWC, LAI, air temperature, relative humidity, and $CO_2$ concentration) that affect the $\Delta^{13}C$ of the summer maize canopy, there was variation in the effects of these factors at different heights within the maize canopy, and the relative contribution of each factor to $\Delta^{13}C$ differed. From the correlation analysis of these various factors to $\Delta^{13}C$, it can be seen that SWC, LAI, and air temperature were the important factors affecting $\Delta^{13}C$ in the lower, middle, and upper layers. However, there was only a weak correlation between relative humidity and the $\Delta^{13}C$ in U, which may be related to the influence of air flow and solar radiation in the upper level of the summer maize canopy. Studies have shown that temperature has a negative correlation with plant $\delta^{13}C$ (*Morecroft & Woodward, 1990*). The results of this experiment demonstrated that temperature was significantly positively correlated with plant $\Delta^{13}C$. Plant growth was largely affected by water availability. There have been many studies on the relationship between water availability or precipitation and the composition of plant $\Delta^{13}C$. For example, *Tambussi, Bort & Araus (2007)* reported that $\Delta^{13}C$ can be used to predict the WUE of $C_3$ plants. *Badeck et al. (2005)* found that different crops or crop organs have different isotopic compositions. The $\Delta^{13}C$ values of different plant organs indicate that their WUE varies. In short, $\Delta^{13}C$ can indirectly indicate crop yield and WUE (*Khan et al., 2007*; *Chen, Chang & Anyia, 2011*; *Gresset et al., 2014*). SWC and LAI have important effects on air temperature, and SWC, LAI, and air temperature are all very significantly

related, indicating that there is some interaction between the three, and mutual adjustment affects the canopy $\Delta^{13}$C. When SWC decreases, crop physiological indicators, including air temperature and stomatal conductance, will first show some characteristics of change. Most of the water absorbed by crops is lost through leaf transpiration to maintain a balanced leaf temperature. If the soil moisture is insufficient, and the evapotranspiration intensity in the air is very large, it will cause the leaf transpiration to decrease and the leaf surface temperature to increase. *Webber et al. (2016)* showed in their study that the air temperature is related to the SWC, and the influence of SWC on crop air temperature in that study was more obvious.

The $\Delta^{13}$C of summer maize is comprehensively controlled by many factors such as SWC, LAI, air temperature, relative humidity, and other factors. Under different field conditions, the contribution of each factor to $\Delta^{13}$C is different, and the combined effects of multiple factors may have a complex interaction with $\Delta^{13}$C. Further studies are required to clearly define the quantitative relationship between factor interactions for a deeper study of the characteristics of $\Delta^{13}$C.

## CONCLUSIONS

Subsoiling and straw return significantly increased the $\Delta^{13}$C of summer maize, and the $\Delta^{13}$C in the middle layer was the smallest among the three layers. The $\Delta^{13}$C of each summer maize layer had a consistent positive correlation with SWC, LAI, and air temperature, and there was a significant positive correlation between the $\Delta^{13}$C in the middle layer (pre-filling stage) and yield. Therefore, carbon isotopes can be used to evaluate summer maize yield, and subsoiling can be used as a reasonable tillage practice to improve summer maize grain yield in the NCP.

## ACKNOWLEDGEMENTS

Special thanks go to the reviewers who had provided much help to improve this paper.

### Funding

This work was supported by the National Nature Science Foundation of China (grant numbers 31771737 and 31471453) and the Special Fund for Agro-scientific Research in the Public Interest of China (grant number 201503117). The funders had no role in study design, data collection and analysis, decision to publish, or preparation of the manuscript.

### Grant Disclosures

The following grant information was disclosed by the authors:
National Nature Science Foundation of China: 31771737 and 31471453.
Public Interest of China: 201503117.

### Competing Interests

The authors declare that they have no competing interests.

## Author Contributions

- Jichao Cui conceived and designed the experiments, performed the experiments, analyzed the data, prepared figures and/or tables, authored or reviewed drafts of the paper, and approved the final draft.
- Huifang Han conceived and designed the experiments, authored or reviewed drafts of the paper, and approved the final draft.

## Data Availability

The raw measurements are available in the Supplemental File.

## Supplemental Information

Supplemental information for this article can be found online at http://dx.doi.org/10.7717/peerj.12891#supplemental-information.

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
