# Peer review of "Carbon isotope discrimination and the factors affecting it in a summer maize field under different tillage systems"

_PeerJ, doi:10.7717/peerj.12891_

## Round 0.1 · original submission · Major Revisions

Dear Authors,

The review process was completed. Both reviewers have advised major modifications before considering your manuscript for publication.
Please, ensure that all requested modifications will be done and provide a point-by-point answer.

Reviewer 1 ·

Basic reporting

1. The manuscript is written in clear language following the PeerJ guidelines but improvement in “Results” section is recommended.
2. Some of the recent articles relevant to their study were not included. Some of the references mentioned in the manuscript were missing from the main text body.
3. Figures and tables are okay. Minor corrections needed.
Details are mentioned in the PDF file.

Experimental design

In some of the methods section, details like the factors studied and the same size for each replicate experiment are missing

Validity of the findings

The authors have tested different parameters in their experiments but have not described them clearly in their results section. It is often misleading to the readers. Some of the methods are very brief and lack details about the experimental factors tested and the number of samples used for each experiment. Hence, the manuscript needs a proper revision before being considered for publishing.

Annotated reviews are not available for download in order to protect the identity of reviewers who chose to remain anonymous.

Reviewer 2 ·

Basic reporting

This study explored the effects of tillage and straw management on carbon isotope discrimination, and the factors affecting carbon isotope during the summer maize crop growing season. The author reported that Δ13C can be used as a prediction index for summer maize yield. It is an interesting and novel study. However, as shown in Fig. 7, a significant correlation was observed between Δ13C and yield at the pre-filling stage, while non-significance at the post-filling stage. Thus, it is suggested that conclusions are rephrased to be more rigorous. Also, the reasons for Δ13C affecting yield should be in-deep discussed. The interaction effects of tillage and straw should be conducted in Table 1 and Figure 3-6.

Specific comments
Line 8-17, the background, objectives, and methods in abstract are redundant.
Line 14-15, upper (U), middle (M), and lower (L) layers. Detailed height.
Line 24-25, should be deleted.
Line 39, reference.
Line 43-46, the author has cleared the crop production in the NCP, but writes the details about maize again. This part should be combined with Line 31.
Line 59, “between WUE, Δ13C, and specific leaf area, the return of …” Confused.
Line 70-76, this part should be placed in M&M. Clear hypothesis was lacked.
Line 90-103, shorten it.
Line 108-120, details of tillage operation could be placed in Supporting Information or as a table.
Line 165-168, Normal distribution was required.
Results: Higher or lower should be based on significance.
Line 212 and 220, delete “at different layers”.
Line 238, the factors affecting it. Detailed factors should be written out.
Line 276-282, It is repeated with Introduction.
Line 286, carbon assimilation capacities affected Δ13C? Reference.
Line 290, “soil respiration emits large quantities of CO2 to the atmosphere, resulting in a lower δ13C on the ground.” Difficult to understand.
Line 321, the relationship between five factors (i.e., SWC, LAI, air temperature, relative humidity, and CO2 concentration) and Δ13C should be discussed in depth. For example, Line 328 and 332, lack references, and explanation was weak.

Experimental design

no comment

Validity of the findings

Please see 1. Basic report

---

## Round 0.2 · Minor Revisions

Dear Authors,

Even that both reviewers have highlighted the improvements done in the manuscript, minor concerns remain. The manuscript should be modified according to the reviewers' comments

Reviewer 1 ·

Basic reporting

The manuscript is written in clear language following the PeerJ guidelines. Raw data files were shared and all the comments made with respect to the literature references, tables and figures were addressed appropriately.

Experimental design

Authors have studied the effect of different factors on Δ13C of summer maize in two growing season. The research design used in the study is appropriate and the methods are clearly mentioned. Few revisions requested by the reviewer were properly addressed and corrections were made in the manuscript.

Validity of the findings

The study performed by the authors is a long experiment done during two growing seasons and includes several parameters. Authors have done their best to represent the data in a reader friendly manner. All the results were clearly presented and missing details in the figures and tables were incorporated in the revised version as requested by the reviewer.

Additional comments

All the corrections were incorporated and the manuscript is revised as per the reviewer's suggestion.

Reviewer 2 ·

Basic reporting

The authors have adressed the concerns. I think it can be accepted after a minor revision. Please double check the whole MS that is free of errors, e.g. superscript and subscript, grammar problems.

Experimental design

no comment

Validity of the findings

no comment/

Additional comments

the subheading of 3.3: it's too long, please shorten it.

---

## Round 0.3 · accepted · Accept

The manuscript was improved by considering all recommendations.